# α-synuclein impairs autophagosome maturation through abnormal actin stabilization

**Souvarish Sarkar**[1], **Abby L. Olsen**[2], **Katja Sygnecka**[1], **Kelly M. Lohr**[1], **Mel B. Feany**[1]*

**1** Department of Pathology, Brigham and Women's Hospital, Harvard Medical School, Boston, Massachusetts, **2** Department of Neurology, Brigham and Women's Hospital, Harvard Medical School, Boston, Massachusetts

* mel_feany@hms.harvard.edu

**Data Availability Statement:** All relevant data are within the manuscript and its Supporting information files.

**Funding:** This work was supported by NIH-NINDS R01-NS098821 and NS0105151 to M.B.F (https://

## Abstract

Vesicular trafficking defects, particularly those in the autophagolysosomal system, have been strongly implicated in the pathogenesis of Parkinson's disease and related α-synucleinopathies. However, mechanisms mediating dysfunction of membrane trafficking remain incompletely understood. Using a *Drosophila* model of α-synuclein neurotoxicity with widespread and robust pathology, we find that human α-synuclein expression impairs autophagic flux in aging adult neurons. Genetic destabilization of the actin cytoskeleton rescues F-actin accumulation, promotes autophagosome clearance, normalizes the autophagolysosomal system, and rescues neurotoxicity in α-synuclein transgenic animals through an Arp2/3 dependent mechanism. Similarly, mitophagosomes accumulate in human α-synuclein-expressing neurons, and reversal of excessive actin stabilization promotes both clearance of these abnormal mitochondria-containing organelles and rescue of mitochondrial dysfunction. These results suggest that Arp2/3 dependent actin cytoskeleton stabilization mediates autophagic and mitophagic dysfunction and implicate failure of autophagosome maturation as a pathological mechanism in Parkinson's disease and related α-synucleinopathies.

## Author summary

Vesicle trafficking is a central cell biological pathway perturbed in Parkinson's disease. Here we use a genetic approach to define an underlying mechanism by demonstrating that the key Parkinson's disease protein α-synuclein impairs maturation of autophagosomes and mitophagosomes through Arp2/3 dependent excess stabilization of cellular actin networks.

## Introduction

Parkinson's disease is the most common neurodegenerative movement disorder and the second most prevalent neurodegenerative disease, after Alzheimer's disease, affecting 1% of

www.nih.gov/). The funders had no role in study design, data collection and analysis, decision to publish, or preparation of the manuscript.

**Competing interests:** The authors have declared that no competing interests exist.

individuals at age 65 [1–3]. Symptoms include motor impairments, as well as nonmotor symptoms. Neuropathologically, Parkinson's disease is characterized by the preferential loss of nigrostriatal dopaminergic neurons and the presence of α-synuclein-rich protein inclusions called Lewy bodies and Lewy neurites. The accumulation of Lewy body inclusions is the shared pathological hallmark of all α-synucleinopathies, a class of neurodegenerative diseases that include Parkinson's disease, dementia with Lewy bodies, and multiple system atrophy [4].

Genetic analysis has provided important insights into Parkinson's disease pathogenesis. In a series of landmark studies, dominant mutations in the gene encoding the synaptic vesicle protein α-synuclein were shown to cause disease, albeit rarely [5–8], making a central connection between protein aggregation, clearance, and disease pathogenesis. Coordinated function of the endolysosomal system is essential to the clearance of misfolded and aggregated proteins during neuronal aging and disease [9]. Both Mendelian and risk loci have implicated altered vesicular trafficking in the pathogenesis of Parkinson's disease and related α-synucleinopathies [10,11]. Mutations in the large multidomain protein LRRK2 are the most common cause of familial Parkinson's disease. Although LRRK2 functions are still being experimentally defined, multiple studies have implicated LRRK2 in controlling autophagy [12–14], perhaps through effects on the actin cytoskeleton [15]. Loci associated with rare monogenic forms of Parkinson's disease, or more complex disorders with a prominent component of parkinsonism, also encode proteins involved in vesicle trafficking: VPS35, ATP13A2, PLA2G6, DNAJC6, SYNJ1, and VPS13C [11,16]. Similarly, loci nominated as risk factors through genome-wide association studies encode proteins, including RAB7L1, SH3GL2, GAK, and CHMP2B, with structural or modulatory roles in vesicle trafficking. In addition, autophagolysosomal dysfunction has been strongly implicated in disease pathogenesis by the substantially increased Parkinson's disease risk in carriers of heterozygous glucocerebrosidase mutations associated with Gaucher's disease [17] and those affected by other lysosomal storage disorders [18,19].

Autophagic degradation specifically of mitochondria has been linked to autosomal recessive Parkinson's disease. Both parkin and PTEN-induced putative kinase 1 (PINK1) play key roles in the turnover of damaged mitochondria through mitophagy [20–22]. Parkin, a cytosolic E3 ubiquitin ligase, translocates from the cytosol to damaged mitochondria following its phosphorylation by PINK1. Parkin then ubiquitinates multiple mitochondrial membrane proteins, which act as substrates for the mitophagy receptors OPTN and NDP52. Autophagy-related (ATG) proteins then assemble and promote the formation of early autophagic structures that isolate damaged mitochondria. Once encapsulated, autophagosomes fuse with lysosomes to degrade vesicle contents.

The precise mechanisms through which α-synuclein pathobiology intersects with altered membrane trafficking in Parkinson's disease and related α-synucleinopathies is unclear. A defect in autophagosomal degradation of α-synuclein [23–26] would plausibly promote the accumulation and deposition of the protein into the insoluble Lewy aggregates characteristic of the α-synucleinopathies. Conversely, multiple studies have suggested that α-synuclein expression leads to membrane trafficking dysfunction in diverse model systems, ranging from yeast to human neurons. In the autophagolysosomal pathway, α-synuclein can impair both macroautophagy (hereafter termed autophagy for simplicity) [27–29] and chaperone-mediated autophagy [29–31]. More generally, α-synuclein expression can impair vesicle trafficking from the endoplasmic reticulum to the Golgi [32–34], with concomitant failure of lysosome hydrolase delivery [35] and endosomal membrane trafficking [32,36–38]. Although these studies collectively strongly support α-synuclein-induced alterations in vesicular membrane trafficking, the mechanistic basis of the interaction is not clear.

In these studies we use a robust new in vivo model of α-synucleinopathy in *Drosophila* [39] to explore the influence of α-synuclein on the autophagolysosomal system. Our model

replicates key features of the human disorder, including age-dependent loss of dopaminergic neurons, structural and functional mitochondrial abnormalities, and extensive α-synuclein aggregation. In addition, we have previously used our new model to demonstrate that α-synuclein binds to spectrin and promotes reorganization of the spectrin cytoskeleton, resulting in excess actin cytoskeleton stabilization, subsequent mislocalization of the critical mitochondrial fission protein Drp1, and mitochondrial enlargement and dysfunction [39,40].

Here we demonstrate that α-synuclein expression promotes morphological and functional abnormalities in the autophagolysosomal system. Specifically, we show impaired autophagosome maturation in neurons expressing transgenic human α-synuclein. Mechanistically, we provide genetic and cellular evidence that α-synuclein-mediated stabilization of the actin cytoskeleton impairs lysosomal fusion of autophagosomes and reduces clearance of both α-synuclein itself and of dysfunctional mitochondria through mitophagy. Altogether, these results suggest that Mendelian and risk loci implicated in α-synucleinopathies act in a mechanistic pathway leading from α-synuclein to mitochondrial dysfunction and dopaminergic neurodegeneration.

## Results

### The autophagolysosomal system is abnormal in α-synuclein transgenic flies

We have recently described a new model of Parkinson's disease and related α-synucleinopathies in the genetic model organism *Drosophila melanogaster* [39]. Our model flies express human wild type α-synuclein specifically in neurons using the *Neurospora*-derived Q system [41] and the pan-neuronal *nSyb-QF2* driver [42]. In addition to age-dependent locomotor dysfunction and progressive neuronal loss, α-synuclein transgenic flies demonstrate robust α-synuclein aggregation, as we have previously reported in a prior *Drosophila* α-synucleinopathy model [43–45]. We determined if α-synuclein expression and aggregation is associated with dysfunction of the autophagolysosomal system by first examining autophagy markers in the α-synuclein transgenic flies. The microtubule-associated protein light chain 3 (LC3) is cleaved, processed, and inserted into nascent autophagosomes, where it is involved in both autophagosome formation and selection of targets for degradation [46]. The autophagy-related gene 8a (Atg8a) protein is the fly homolog of human LC3 and is widely used to mark autophagic structures in *Drosophila*. We detected autophagosomes by crossing flies containing the transgenic reporter construct *UAS-Atg8a-GFP* to α-synuclein transgenic flies and visualized tagged Atg8a by immunofluorescence in brain sections. Human α-synuclein expression increased the number of GFP-positive puncta in α-synuclein transgenic fly brains compared to age-matched control animals (Fig 1A arrows and Fig 1D) in an age-dependent manner (S1A Fig arrows and S1B Fig). We further confirmed these results for endogenous Atg8a using an antibody recognizing the protein (Fig 1B arrows and Fig 1D and S1C Fig arrows and S1D Fig) [46–48]. All quantitative analysis of puncta was performed in the anterior medulla (S2A Fig), a site of prominent pathology and strong transgene expression in the current α-synuclein transgenic model [39].

The p62 protein, also known as sequestosome 1 (SQSTM1), selectively mediates degradation of ubiquitinated proteins and protein aggregates by binding to both ubiquitin and LC3. Upon autophagic activation, p62 is recruited to autophagosomes and eventually degraded in lysosomes. p62 accumulation frequently accompanies aberrant autophagosome maturation and is commonly used as a measurement of autophagic flux [46,49]. The *Drosophila* homolog of p62, ref(2)P, is a component of protein aggregates formed in brain and peripheral tissues under conditions of disrupted autophagy, including neurodegenerative disease and aging [50]. Immunofluorescence with an antibody directed to *Drosophila* p62 revealed an increase in

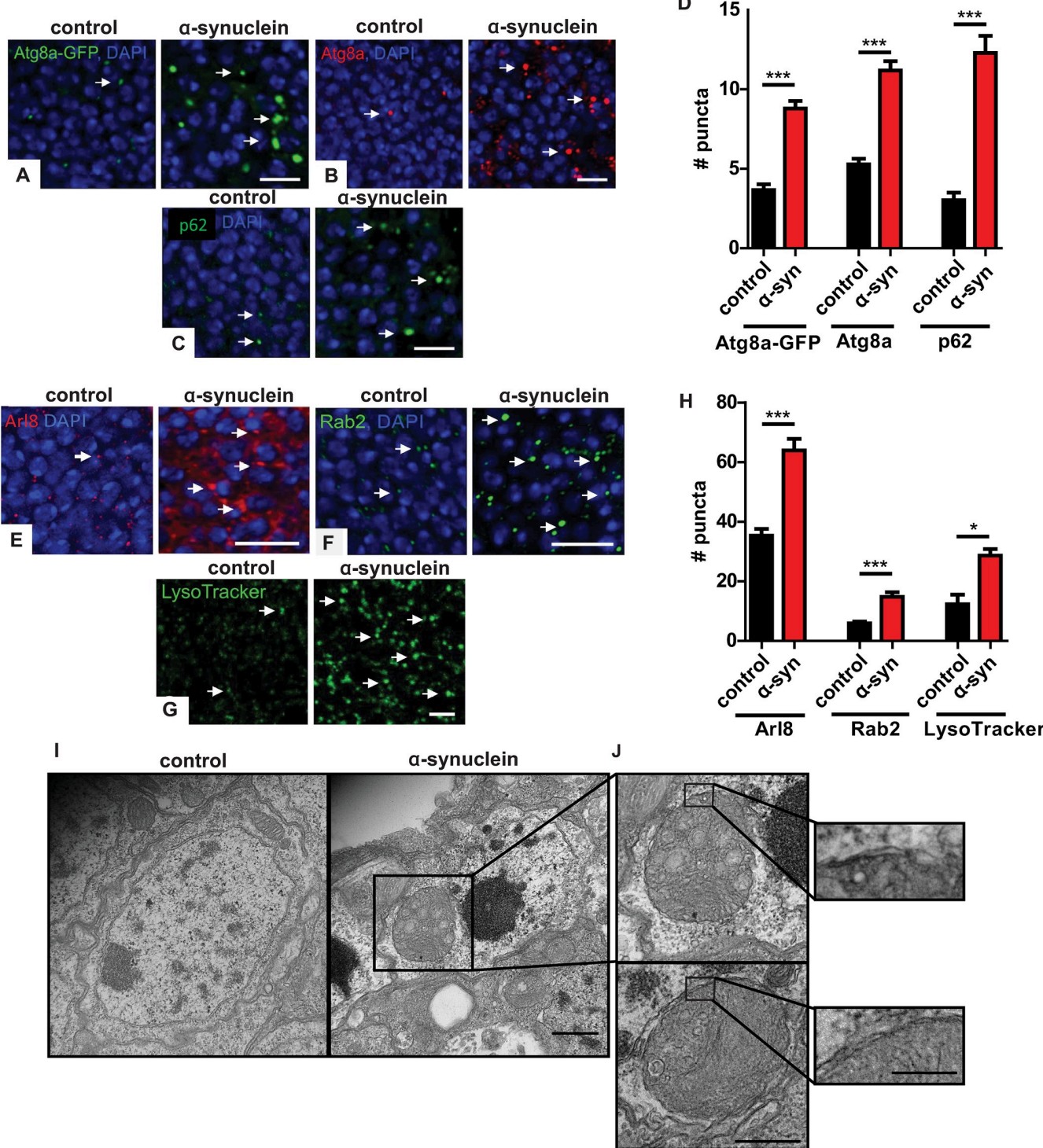

**Fig 1. α-synuclein transgenic fly brains show increased autophagosome and lysosome markers.** (A) Increased autophagosomes in α-synuclein transgenic flies visualized with the Atg8a-GFP reporter. Arrows indicate GFP-positive puncta. (B) Increased autophagosomes in α-synuclein transgenic flies visualized with an antibody recognizing endogenous Atg8a. Arrows indicate Atg8a-positive puncta. (C) Increased p62-positive puncta in brains of α-synuclein transgenic flies. Arrows indicate p62-positive puncta. (D) Quantitative analysis of the number of puncta in α-synuclein and control flies. (E) Increased immunofluorescence for the lysosomal marker Arl8 in brains of α-synuclein transgenic flies. Arrows indicate Arl8-positive puncta. (F) Increased lysosomes visualized via GFP staining of Rab2<sup>EYFP</sup> in α-synuclein transgenic fly brains. Arrows indicate Rab2-positive lysosomes. (G) Increased lysosomes labeled with LysoTracker Green from whole mount brains of α-synuclein transgenic flies compared to controls. Arrows indicate LysoTracker Green-positive puncta. (H)

Quantitative analysis of the number of lysosomes in α-synuclein and control flies. (I-J) Transmission electron microscopy reveals abnormal, enlarged autophagosomes in α-synuclein transgenic, but not control, flies. Insets demonstrate enlarged autophagosomes and double membranes bounding the autophagosomes. Control genotype in (A): *UAS-Atg8a-GFP/ nSyb-QF2, nSyb-GAL4*. Control genotype in (B-E, G-J): *nSyb-QF2, nSyb-GAL4/+*. Control genotype in (F): *Rab2^{EYFP}/+; nSyb-QF2, nSyb-GAL4/+*. Full genotypes are provided for all animals in S1 Text. * p<0.05, *** p<0.001, Student's t-test. Data are represented as mean ± SEM. n = 6 per genotype. Scale bars are 5 µm in (A-C,E-G), 500 nm in (I) 400 nm in (J) and 250 nm in (J), inset. The number of puncta per 500 µm$^2$ is presented in (A-H). Flies are 10 days old in (A-H) and 20 days old in (I,J).

p62-positive puncta in brains of α-synuclein transgenic flies compared to controls, suggesting α-synuclein-mediated impairment of autophagy (Fig 1C arrows and Fig 1D). The effects of α-synuclein on p62-positive puncta were also age-dependent (S1E and S1F Fig).

Expansion of the lysosomal compartment frequently accompanies abnormalities in autophagy. We first assessed lysosomes in α-synuclein transgenic flies by immunostaining α-synuclein transgenic and control fly brains for Arl8, the *Drosophila* homolog of mammalian Arl8a and Arl8b, Arf-like GTPases localized to lysosomal membranes, which control lysosomal motility [51,52]. Analysis of Arl8 staining revealed an increase in the number of lysosomes in the brains of α-synuclein transgenic flies compared to controls (Fig 1E arrows and Fig 1H). Rab GTPases are a family of proteins involved in the regulation of intracellular membrane trafficking, with different members localizing to specific membrane compartments [53]. Rab2 plays a role in autophagic degradation in lysosomes [54]. Using a *Drosophila* protein trap line in which Rab2^{EYFP} is expressed under the control of endogenous Rab2 regulatory sequences [55], we showed an increase in Rab2-positive puncta in α-synuclein transgenic fly brains compared to controls (Fig 1F arrows and Fig 1H). We assessed the ability of lysosomes to acidify by labeling freshly dissected brains with LysoTracker Green. LysoTracker is a membrane-permeable dye, which selectively stains acidic compartments, including lysosomes. Increased numbers of LysoTracker-positive puncta were present in the brains of α-synuclein flies compared to controls (Fig 1G arrows and Fig 1H) consistent with our results with Arl8 and Rab2 immunostaining and suggesting that lysosomal acidification occurs in α-synuclein transgenic fly brains. Compared to Rab2 and LysoTracker staining, Arl8 immunostaining appeared more diffuse in α-synuclein transgenic fly brains (Fig 1E), which may reflect the presence of Arl8 in late endosomes as well as lysosomes [51,56]. Additional investigation will be required to test this hypothesis. Autophagosome and lysosome abnormalities do not appear to simply reflect expression of an exogenous protein because we do not observe changes in Atg8a, p62 or Arl8 in flies expressing EGFP or ß-galactosidase (S2B–S2G Fig).

We next examined cellular morphology of α-synuclein transgenic flies in more detail using transmission electron microscopy. Numerous large, double membrane-bound cytoplasmic organelles filled with membranous and vesicular cellular components, consistent with enlarged autophagosomes, were present in neurons from human α-synuclein transgenic flies but not in control fly neurons (Fig 1I and 1J insets).

## Autophagic flux is impaired in α-synuclein-expressing neurons

The abnormalities in autophagosomal and lysosomal markers and morphology suggest significant dysregulation of the autophagolysosomal system in α-synuclein transgenic flies. Based on the increased p62 immunostaining (Fig 1C and 1D), we next examined whether α-synuclein transgenic neurons showed impaired autophagic flux. We examined autophagic flux directly by expressing a GFP-mCherry-Atg8a reporter in neurons of α-synuclein transgenic and control flies. The tandem fluorescent GFP-mCherry-Atg8a protein has been used to follow the maturation and progression of autophagosomes to autolysosomes in a variety of model systems [49,57], including *Drosophila* [46,58]. When the GFP-mCherry-Atg8a protein is in the acidic autolysosome, GFP fluorescence is quenched, leaving only the mCherry signal. Thus, a

small GFP to mCherry ratio indicates preserved autophagic flux, while a larger ratio indicates a flux deficit. The brains of α-synuclein transgenic flies expressing the GFP-mCherry-Atg8a reporter displayed increased GFP-positive puncta that were also positive for mCherry (Fig 2A arrows and Fig 2B), as quantified by a higher GFP to mCherry ratio (Fig 2D and 2E). These findings are consistent with the presence of autophagosomes and reduced autolysosomal formation, indicating impaired autophagic flux in α-synuclein transgenic neurons. Again, our results appear specific because expression of an unrelated protein, ß-galactosidase, does not perturb autophagic flux (S2H and S2I Fig).

## Genetic destabilization of F-actin normalizes autophagy and mitophagy in α-synuclein transgenic flies

We next addressed the mechanism by which increased levels of neuronal α-synuclein alter the morphology and function of the autophagolysosomal system. We have previously demonstrated that α-synuclein expression results in excess stabilization of the actin cytoskeleton through the actin binding protein spectrin, as reviewed in detail above [39]. We assessed the role of the actin cytoskeleton in the autophagolysosomal abnormalities present in α-synuclein transgenic flies by overexpressing the actin severing proteins gelsolin or cofilin (*Drosophila* twinstar). We have previously documented reduced F-actin levels in the brain of flies expressing either transgenic gelsolin [40] or cofilin [59]. Gelsolin or cofilin expression resulted in significant rescue of the α-synuclein-mediated increase in large Atg8a-GFP puncta greater than 0.5 μm in diameter (Fig 3A arrows and Fig 3B), p62 aggregates (Fig 3C arrows and Fig 3D), Arl8-positive lysosomes (Fig 3E arrows and Fig 3F), and Lysotracker-labeled lysosomes (Fig 3G arrows and Fig 3H). Expression of cofilin with the inhibitory S3E mutation [60–62] did not rescue the effect of α-synuclein on the number of Atg8a-GFP-positive puncta (S3A and S3B Fig), despite equivalent levels of cofilin expression (S3D Fig). Interestingly, we saw a specific decrease in large Atg8a-GFP puncta greater than 0.5 μm in diameter, but not smaller puncta, with overexpression of gelsolin or cofilin. The mechanistic explanation of the specific effect on large puncta will require further investigation. We ensured that normalization of autophagosome and lysosome number and morphology did not simply reflect decreased levels of α-synuclein by performing western blots, which revealed no change in α-synuclein protein levels with expression of gelsolin or cofilin (S3C and S3E Fig). Since α-synuclein is expressed using the Q system [41] while gelsolin and cofilin are expressed using the noninteracting GAL4/UAS system [63], we do not expect alterations in transgene expression based on transcriptional dilution effects. Gelsolin or cofilin overexpression in the absence of transgenic α-synuclein did not alter markers of autophagosomes or lysosomes (S4A–S4H Fig).

We next determined the effect of genetic F-actin cytoskeleton destabilization by gelsolin or cofilin overexpression on autophagic flux using the GFP-mCherry-Atg8a reporter. Gelsolin or cofilin overexpression significantly reduced the numbers of GFP- and mCherry-positive puncta in α-synuclein flies (Fig 4A arrows and Fig 4B–4D). The number of dual GFP and mCherry positive puncta in flies expressing cofilin or gelsolin was significantly decreased as indicated by counting the number of double-positive puncta and quantifying the GFP to mCherry ratio (Fig 4E and 4F). These findings are consistent with a significant rescue of autophagic flux by F-actin destabilization in α-synuclein transgenic flies. We found no effect of gelsolin or cofilin overexpression on autophagic flux in control flies not expressing human α-synuclein (S4I and S4J Fig).

Abnormalities in mitophagy have been strongly implicated in the pathogenesis of Parkinson's disease [4,22]. Since mitophagy relies on the autophagosomal degradation of mitochondria, we next determined if abnormal autophagosomes in α-synuclein transgenic neurons

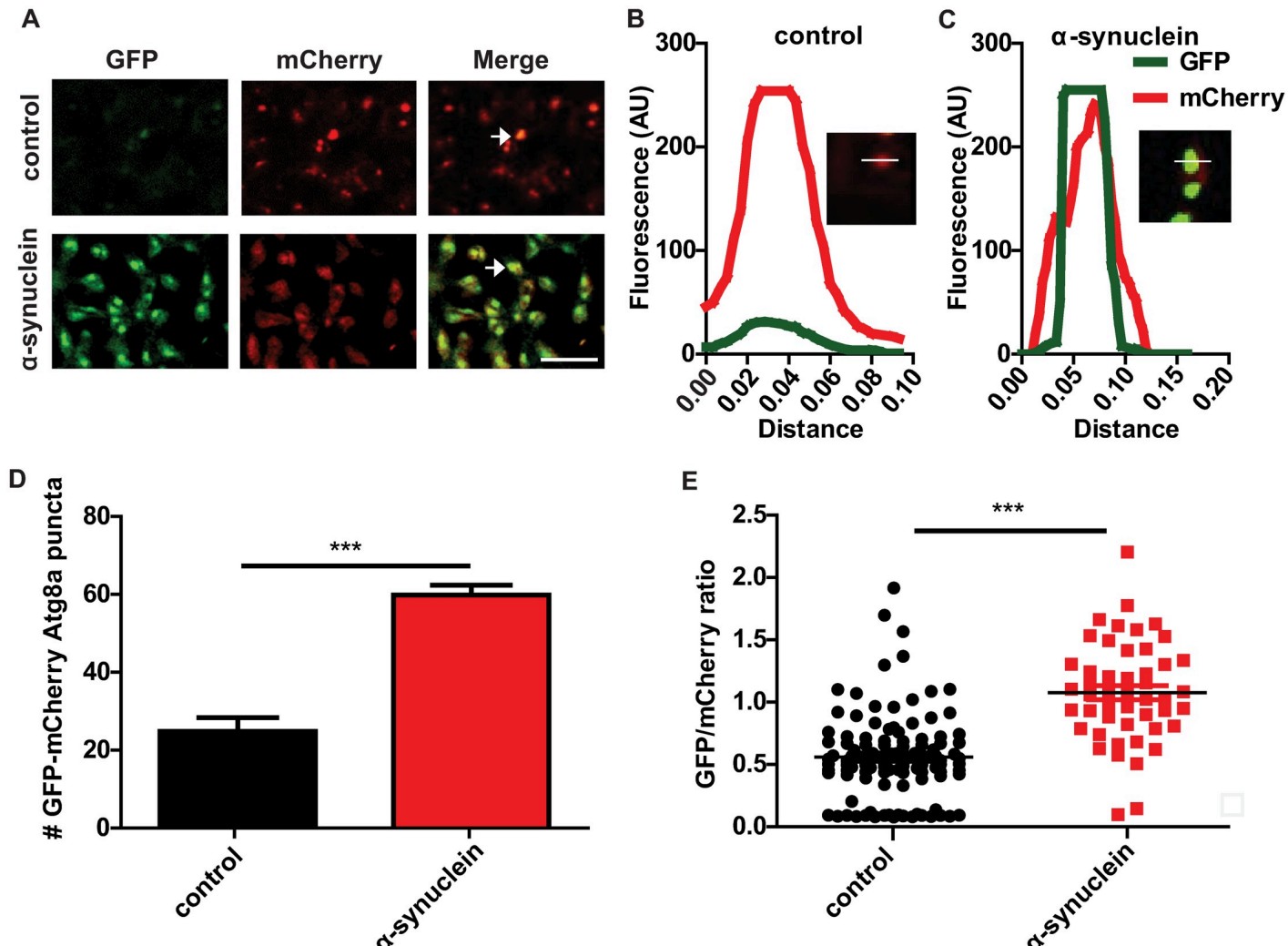

**Fig 2. α-synuclein transgenic fly brains show impaired autophagic flux.** (A) Representative images of brains from control and α-synuclein transgenic flies also expressing GFP-mCherry-Atg8a. Arrows indicate GFP and mCherry double-positive puncta. (B-C) Fluorescence intensity plot across a region of interest in control and α-synuclein transgenic flies. Inset indicates region plotted. (D) Increased GFP-mCherry-Atg8a dual-positive puncta in α-synuclein transgenic flies. (E) Increased ratio of GFP to mCherry fluorescence in brains of α-synuclein transgenic flies. Control genotype: *UAS-GFP-mCherry-Atg8a/+; nSyb-QF2, nSyb-GAL4/+*. Full genotypes are provided for all animals in S1 Text. *** p<0.001, Student's t-test. Data are represented as mean ± SEM. n = 3 per genotype. Scale bar is 5 μm (A). The number of puncta per 1000 μm$^2$ is presented. Flies are 10 days old.

contained mitochondria. We used Airyscan super-resolution imaging of sections immunolabeled using an antibody directed to ATP5A to mark mitochondria [39,40,64] and mCherrry-Atg8a to identify autophagosomes. α-synuclein transgenic fly brains showed colocalization of abnormal, enlarged autophagosomes with mitochondria (Fig 5A arrows and Fig 5B and S1 Video). In contrast, control fly brains showed significantly less autophagosome and mitochondrial colocalization.

We examined mitophagy more directly in our α-synuclein transgenic flies using the mt-Keima transgenic reporter, which monitors PINK1/parkin-mediated mitophagy [65]. The excitation spectrum of the mt-Keima fluorescent protein shifts when it is delivered to the acidic lysosome, which can be detected by dual-excitation ratiometric quantification. When the mt-Keima reporter was expressed in α-synuclein transgenic fly neurons in vivo, we

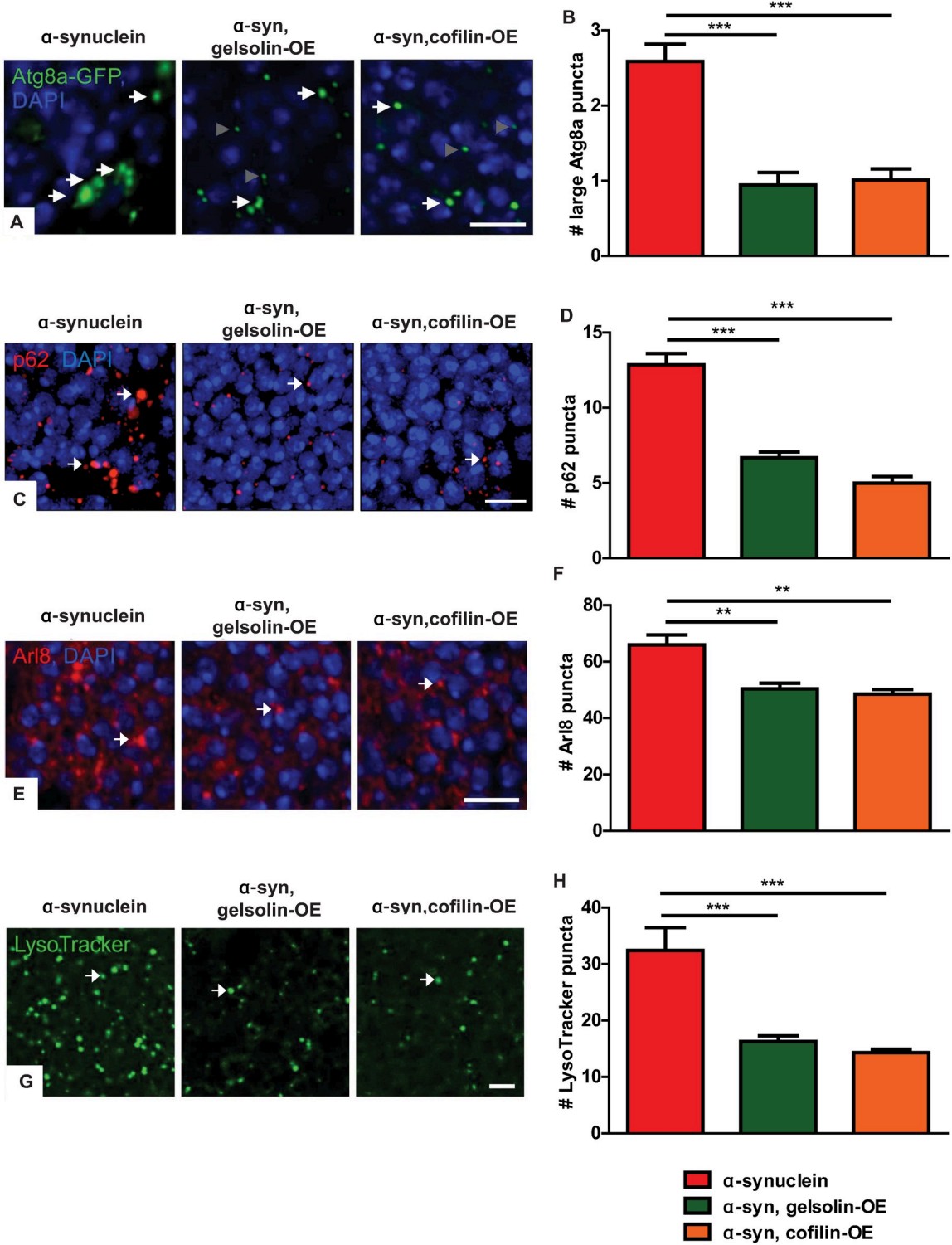

**Fig 3. Gelsolin or cofilin overexpression rescues autophagosome and lysosome marker abnormalities in α-synuclein transgenic fly brains.** (A) Representative images of GFP immunofluorescence in α-synuclein transgenic flies expressing Atg8a-GFP with and without gelsolin or cofilin overexpression. White arrows indicate large Atg8a-GFP-positive puncta included in quantification, and gray arrowheads indicate smaller puncta not included in quantification. (B) Quantification of large Atg8a-GFP-positive puncta (>0.5 μm in diameter). (C) Representative images of p62-positive puncta (arrows) in α-synuclein transgenic flies with and without gelsolin or cofilin overexpression. (D) Quantification of p62-positive puncta. (E) Representative images of lysosomes (arrows) highlighted by

immunostaining for the lysosomal marker Arl8 in α-synuclein transgenic flies with and without gelsolin or cofilin overexpression. (F) Quantification of Arl8-positive puncta. (G) Representative images of lysosomes identified with LysoTracker Green staining of freshly dissected brains (arrows) in α-synuclein transgenic flies with and without gelsolin or cofilin overexpression. (H) Quantification of LysoTracker-positive puncta. α-synuclein genotype in (A-B): *QUAS-wild type α-synuclein, nSyb-QF2, nSyb-GAL4, UAS-Atg8a-GFP / +*. α-synuclein genotype in (C-H): *QUAS-wild type α-synuclein, nSyb-QF2, nSyb-GAL4 / +*. Full genotypes are provided for all animals in S1 Text. ** $p < 0.01$, *** $p < 0.001$, ANOVA with Tukey's multiple comparisons test. Data are represented as mean ± SEM. n = 6 per genotype. Scale bars are 5 μm. The number of puncta per 500 μm$^2$ is presented. Flies are 10 days old.

observed a significant decrease in the 543 nm to 458 nm ratio, consistent with reduced delivery of mitochondria to an acidic compartment for degradation (Fig 5C arrows and Fig 5D). Notably, overexpression of either gelsolin or cofilin significantly restored mitophagy as assessed by the mt-Keima reporter (Fig 5E arrows and Fig 5F). Gelsolin or cofilin overexpression with the transgenes used in the current study had no effect on mitophagy in the absence of transgenic human α-synuclein expression (S5 Fig).

To determine if normalization of mitophagy was accompanied by improved mitochondrial function, we employed a recently described method for assessment of metabolism in whole fly brains using the Agilent Seahorse XFe96 Analyzer [66], which we applied to our α-synuclein transgenic animals. Expression of human α-synuclein in neurons reduced brain oxygen consumption rate (OCR, Fig 6A), extracellular acidification rate (ECAR, Fig 6B), proton efflux rate (PER, Fig 6C), non-mitochondrial respiration (Fig 6D), basal respiration (Fig 6E), maximal respiration (Fig 6F), and proton leak (Fig 6G). Gelsolin or cofilin overexpression significantly improved each of these metabolic parameters, consistent with normalization of downstream pathways of neurotoxicity by F-actin stabilization. Increased expression of gelsolin had no effect on metabolic parameters in the absence of transgenic human α-synuclein expression. Cofilin overexpression with the transgene used in these studies modestly elevated basal respiration and decreased non-mitochondrial respiration in the absence of transgenic human α-synuclein expression (S6A–S6G Fig).

## Genetic destabilization of the actin cytoskeleton rescues α-synuclein-mediated neurotoxicity and aggregation

We probed the effects of F-actin destabilization on α-synuclein-mediated neurotoxicity more directly by assessing the total number of neurons on histological sections of the anterior medulla [39]. Gelsolin and cofilin overexpression rescued the α-synuclein-mediated reduction in total neuron number (Fig 7A). Next, we determined the effect of gelsolin and cofilin overexpression specifically on degeneration of dopaminergic neurons using tyrosine hydroxylase immunostaining with a monoclonal antibody, as previously described [39,45,67]. Gelsolin and cofilin overexpression significantly rescued the α-synuclein-mediated loss of tyrosine hydroxylase-positive dopaminergic neurons (Fig 7B arrows and Fig 7C). Since autophagy can mediate clearance of abnormal protein aggregates, including α-synuclein inclusions [26,68], we next examined α-synuclein protein aggregation in our α-synuclein transgenic flies following genetic destabilization of the actin cytoskeleton. Gelsolin or cofilin overexpression significantly decreased neuronal α-synuclein aggregates in α-synuclein transgenic flies as shown by immunostaining (Fig 7D arrows and Fig 7E).

## The Arp2/3 complex controls autophagolysosomal pathology in α-synuclein transgenic flies

We next performed a candidate-based screen of actin regulatory factors to probe the mechanisms by which α-synuclein promotes stability of the actin cytoskeleton and impairs

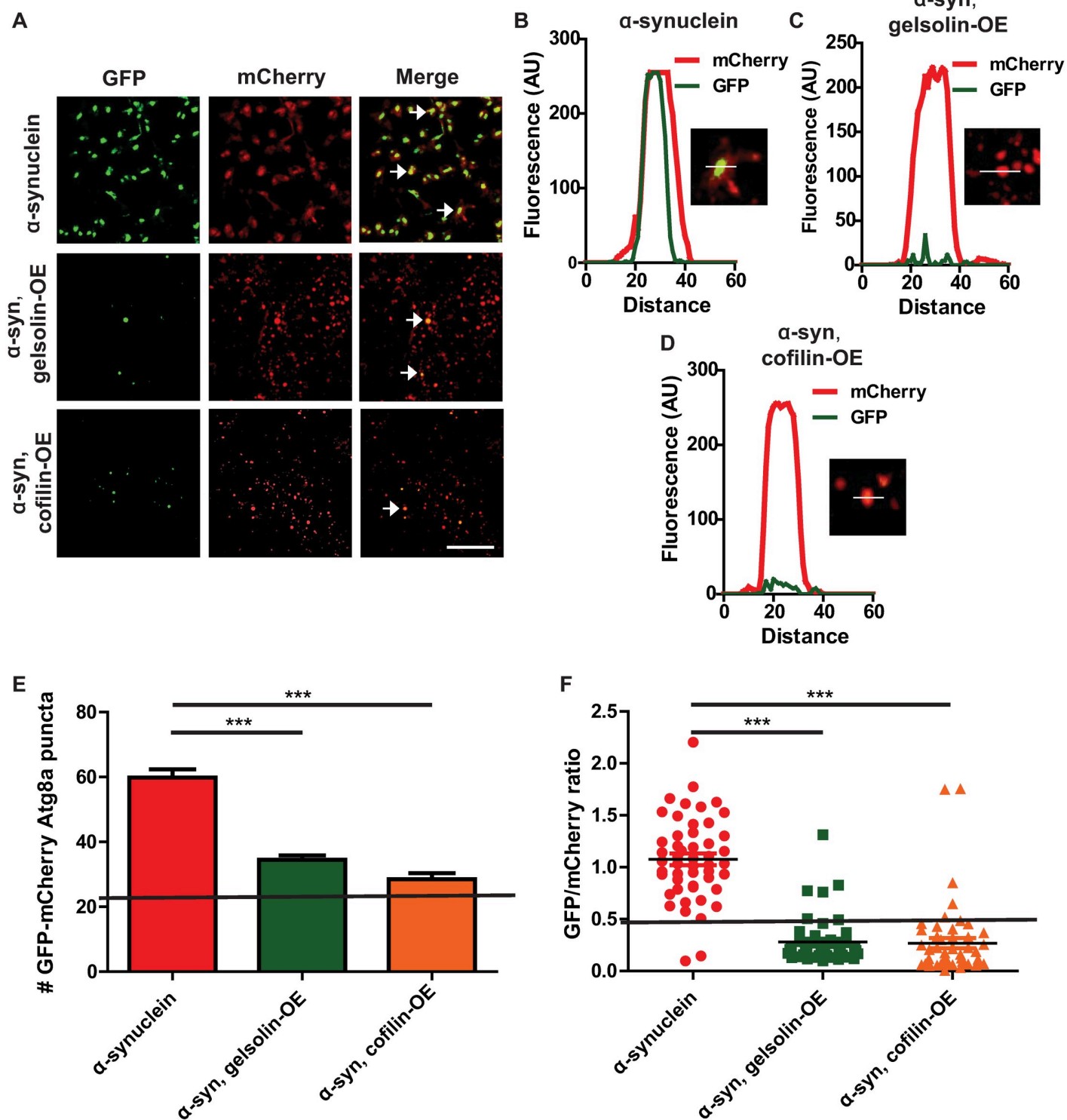

**Fig 4. Gelsolin or cofilin overexpression restores autophagic flux in α-synuclein transgenic flies.** (A) Representative super-resolution brain images from control and α-synuclein transgenic flies expressing GFP-mCherry-Atg8a. Arrows indicate GFP and mCherry double-positive puncta. (B-D) Fluorescence intensity plot across regions of interest in α-synuclein transgenic flies with and without gelsolin or cofilin overexpression. Insets indicate regions plotted. (E) Total number of GFP-mCherry-Atg8a-positive puncta in α-synuclein transgenic flies with and without gelsolin or cofilin overexpression. (F) Ratio of GFP to mCherry fluorescence in brains α-synuclein transgenic flies with and without gelsolin or cofilin overexpression. The solid lines in (E,F) indicate values for controls (genotype: *UAS-GFP-mCherry-Atg8a/+; nSyb-QF2, nSyb-GAL4/ +*), see S4I and S4J Fig. α-synuclein genotype: *UAS-GFP-mCherry-Atg8a/+; QUAS-α-synuclein / nSyb-QF2, nSyb-GAL4*. Full genotypes are provided for all animals in S1 Text. *** p<0.001, ANOVA with Tukey's multiple comparisons test. Data are represented as mean ± SEM. Scale bar is 5 μm (A). n = 4 per genotype. The number of puncta per 1000 μm$^2$ is presented. Flies are 10 days old.

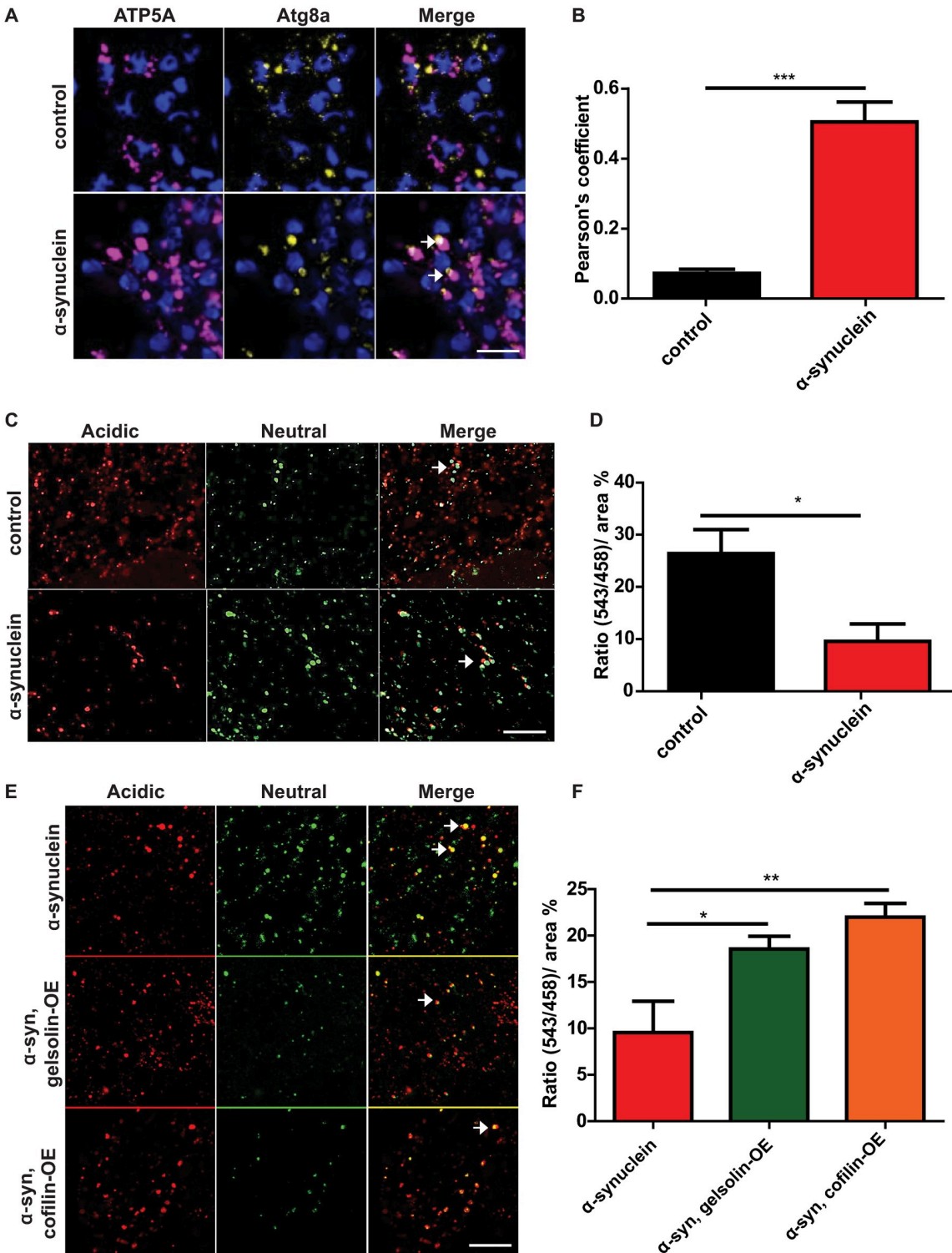

**Fig 5. Gelsolin or cofilin overexpression rescues abnormal mitophagy in α-synuclein transgenic flies.** (A) Representative super-resolution images of brains from control and α-synuclein transgenic flies showing colocalization of mitochondria (ATP5A) with enlarged autophagosomes marked by transgenic mCherry-Atg8a (arrows). See also S1 Video. (B) Quantification of colocalization in A. (C) Overlay of live mt-Keima emission at 543 (red) and 458 (green) showing fewer high 543/458 ratio puncta in α-synuclein transgenic brains. (D) High (543/458) ratio area/total mitochondrial area was quantified as an index of mitophagy in control and α-synuclein transgenic flies and shows a reduction with human α-synuclein expression. (E). Overlay of live mt-Keima emission at 543 (red) and 458

 

(green) in α-synuclein transgenic brains also overexpressing either gelsolin or cofilin. (F) 543/458 ratio area/total mitochondrial area was quantified as an index of mitophagy in control and α-synuclein transgenic flies and shows a rescue of mitophagy in α-synuclein transgenics by of gelsolin or cofilin overexpression. Control genotype in (A,B): *UAS-GFP-mCherry-Atg8a / +; nSyb-QF2, nSyb-GAL4 / +.* Control genotype in (C-F): *UAS-mt-Keima/ +; nSyb-QF2, nSyb-GAL4/ +.* Full genotypes are provided for all animals in S1 Text. * p<0.05, ** p<0.01, ANOVA with Tukey's multiple comparisons test. Data are represented as mean ± SEM. n = 4 per genotype. Scale bars are 5 μm in (A) and 10 μm in (B,D). Flies are 10 days old.

autophagy. Knocking down members of the Arp2/3 actin nucleating complex, including Arp2, Arp3, and Arpc1, using transgenic RNAi produced robust and reliable decreases in Atg8a, p62, and Arl8 puncta in α-synuclein transgenic fly brains (Fig 8 arrows and S7A Fig arrows and S7B Fig). The modifying effect of these genes did not reflect decreased transgenic α-synuclein, as determined by western blot analysis (S7C Fig). The partial loss of function reagents tested here had no effect on the modest number of Atg8a or p62 puncta present in control flies (Figs 8A–8D and S7B Fig). However, knockdown of Arp2, Arp3, or Arpc1 did reduce the number of lysosomes present in otherwise wild type flies (Fig 8E arrows and Fig 8F), consistent with a role for the Arp2/3 complex in controlling lysosome homeostasis in otherwise wild type flies.

## Discussion

Increased expression of autophagic and lysosomal markers have been well documented in postmortem tissue from Parkinson's disease and related α-synucleinopathies [68–72]. Based on α-synuclein's hypothesized toxic gain of function and the high penetrance of familial Parkinson's disease in patients with α-synuclein locus duplications and triplications [4,8], multiple models of α-synuclein aggregation and neurotoxicity have been created by expressing human wild type α-synuclein in rodents. Brains from these rodent models also show induction of autophagic and lysosomal markers [68,70,72], further connecting α-synuclein with alterations in the autophagolysosomal system. However, the molecular mechanisms linking the two have not been fully defined. We have recently described a new model of α-synucleinopathy in *Drosophila* based on neuronal expression of human wild type α-synuclein using the pan-neuronal *nSyb-QF2* driver. Our model demonstrates widespread age-dependent accumulation of α-synuclein aggregates and progressive neurodegeneration [39]. We show here that α-synuclein transgenic flies show increased markers of autophagosomes and lysosomes (Fig 1), similar to that seen in human α-synucleinopathies and rodent models. We further use our new model to demonstrate a key role for abnormal F-actin stabilization in promoting the autophagolysosomal dysfunction induced by human wild type α-synuclein.

We have previously demonstrated that α-synuclein interacts with the actin binding protein spectrin to mediate downstream neurotoxicity by abnormally stabilizing the F-actin cytoskeleton [39]. Here we have confirmed and extended these studies by showing that overexpression of either of the actin severing proteins gelsolin [40,73] or cofilin [59,74] protects from α-synuclein-induced neurodegeneration (Fig 7). We additionally find that genetic destabilization of the actin cytoskeleton reverses autophagosomal dysfunction (Figs 3 and 4), while the Arp2/3 actin nucleating complex family members Arp2, Arp3 and Arpc1 promote autophagolysosomal abnormalities in α-synuclein transgenic animals (Fig 8). In our in vivo system, α-synuclein expression also impairs autophagic flux (Fig 2), which is accompanied by an accumulation of abnormal autophagosomes. These findings are consistent with prior cell culture work demonstrating that actin dynamics regulates lysosome-autophagosome fusion [75,76], including by promoting autophagosome trafficking to the lysosome [77]. Actin also participates in other aspects of autophagy [78,79] that may also or alternatively contribute to the F-actin mediated defects in autophagic flux observed in our α-synuclein transgenic flies. During

 

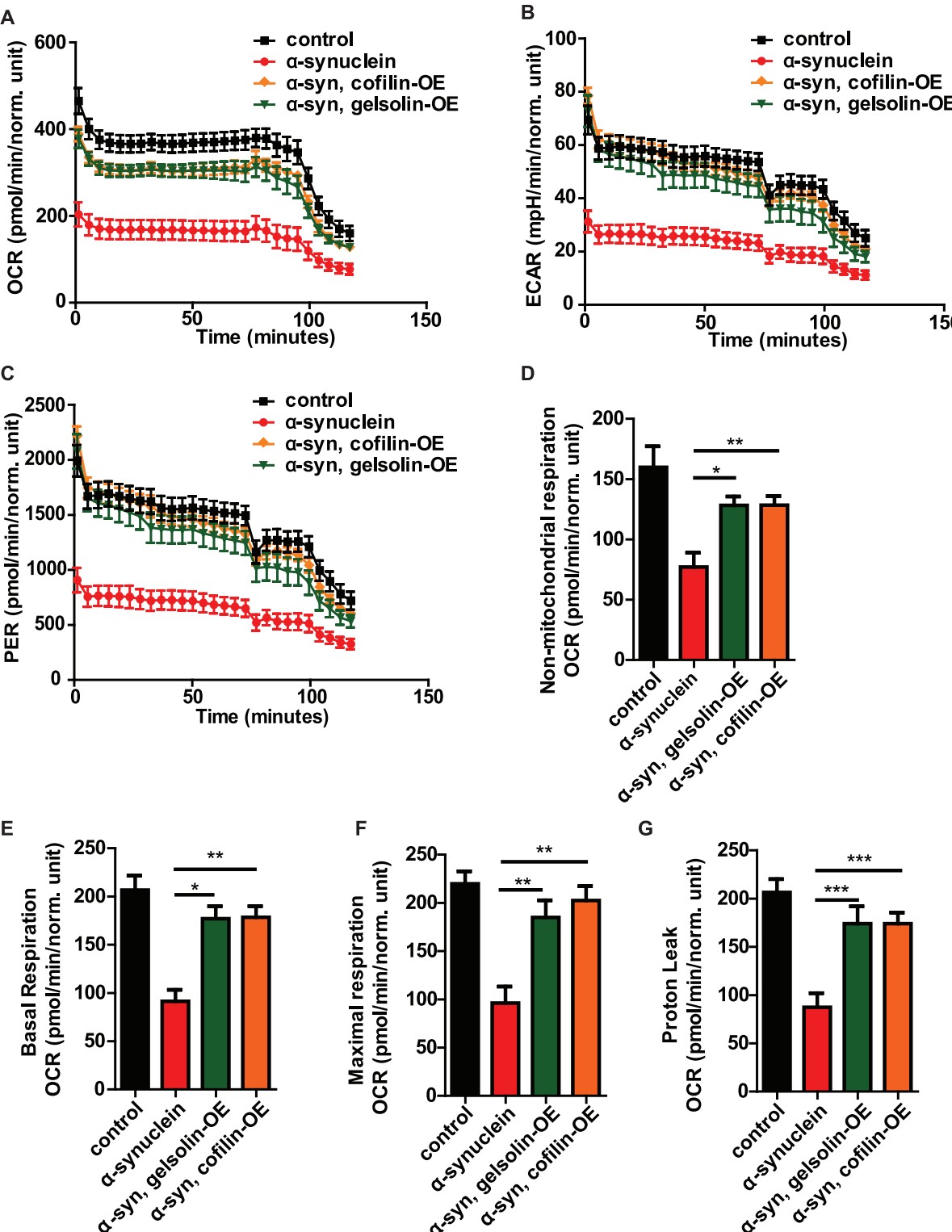

**Fig 6. Gelsolin or cofilin overexpression normalizes bioenergetics in α-synuclein transgenic flies.** (A-G) Metabolic profiling of whole *Drosophila* brains in Seahorse XFe96-well culture microplates (Agilent) reveals decreased oxygen consumption rate (OCR, A), extracellular acidification rate (ECAR, B), proton efflux rate (PER, C), non-mitochondrial respiration (D), basal respiration (E), maximal respiration (F), and proton leak (G) mediated by α-synuclein expression, which is substantially rescued by gelsolin or cofilin overexpression. Control genotype: *nSyb-QF2, nSyb-GAL4/+*. Full genotypes are provided for all animals in S1 Text. * p<0.05, ** p<0.01, *** p<0.001, ANOVA with Tukey's multiple comparisons test. Data are represented as mean ± SEM. n = 6 per genotype. Flies are 10 days old.

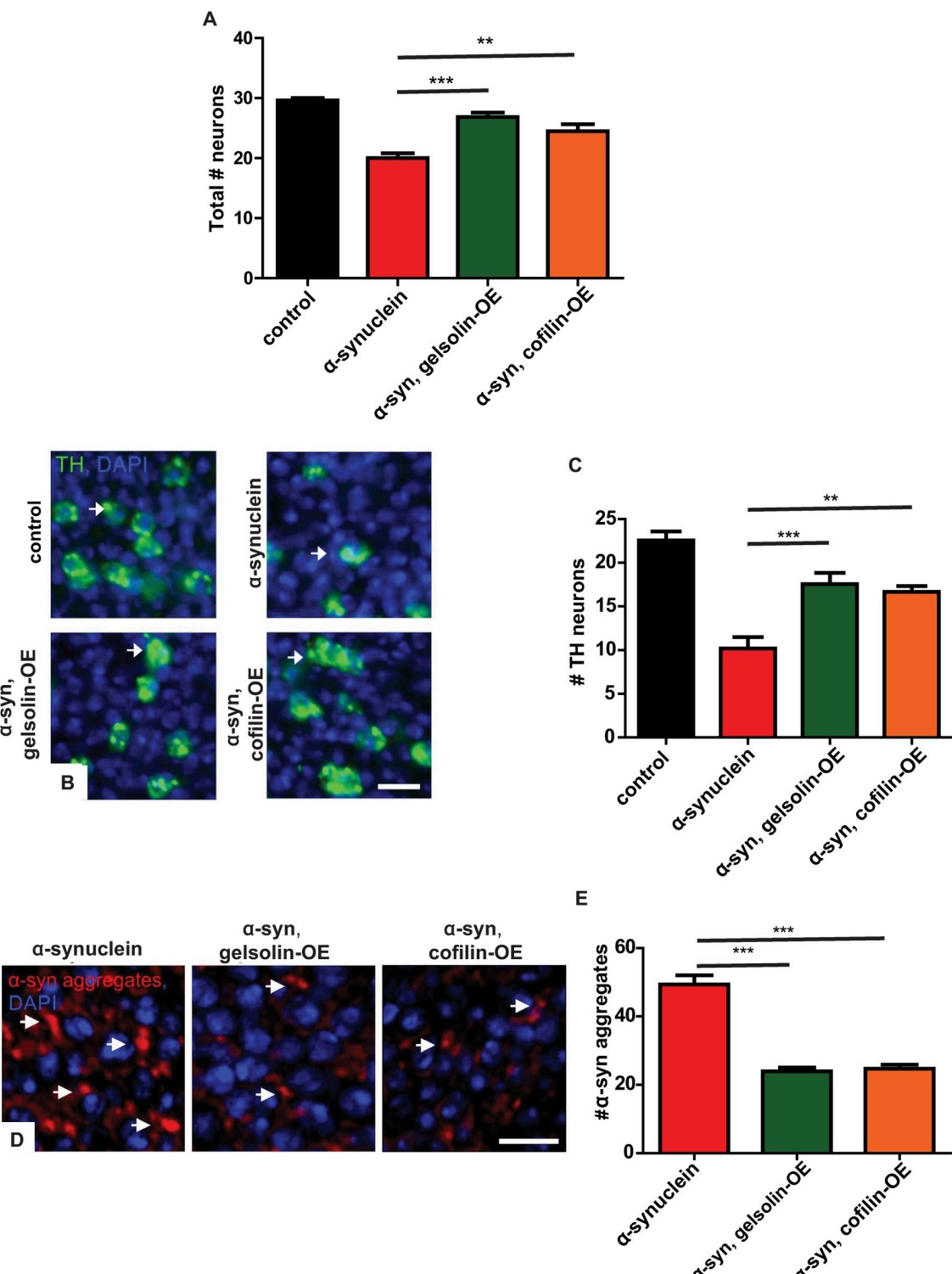

**Fig 7. Gelsolin or cofilin overexpression rescues neurodegeneration and decreases inclusions in α-synuclein transgenic flies.** (A) Neuronal loss in the anterior medulla of α-synuclein transgenic flies is significantly rescued by overexpression of either gelsolin or cofilin. (B) Representative images of tyrosine hydroxylase (TH)-positive dopamine neurons (arrows) in the medulla of α-synuclein transgenic flies at baseline and with gelsolin or cofilin overexpression. (C) Quantification of the number of tyrosine hydroxylase-positive neurons. (D) Representative images of α-synuclein immunoreactive inclusions (arrows) from α-synuclein transgenic flies at baseline and with gelsolin or cofilin overexpression. (E) Quantification of α-synuclein aggregates. Control genotype: *nSyb-QF2, nSyb-*

*GAL4/+*. Full genotypes are provided for all animals in S1 Text. ** p<0.01, *** p<0.001, ANOVA with Tukey's multiple comparisons test. Data are represented as mean ± SEM. n = 6 per genotype. Scale bars are 5 μm. The number of TH-positive neurons per 10,000 μm$^2$ is presented (C). The number of aggregates per 200 μm$^2$ is presented (E). Flies are 10 days old.

autophagosome formation a branched actin network shapes the developing omegasome and phagopore [80]. Additionally, actin comet tail motility promotes formation of the autophagosome from the phagopore [77,81].

Normal actin cytoskeletal dynamics have been specifically implicated in autophagosome-lysosome fusion during selective autophagy [76]. We find that one form of selective autophagy, mitophagy, is impaired in our α-synuclein transgenic flies (Fig 5), with consequent defects in cellular bioenergetics (Fig 6). As discussed, our data implicate F-actin-dependent failure of autophagosome maturation and improper degradation of autophagosome contents, including mitochondria (Fig 5). However, proper actin dynamics are also critical for other aspects of mitophagy, including formation of protective F-actin around damaged mitochondria [82] and disassembly of mitochondrial aggregates [83] prior to mitophagy. In addition, we have previously shown that impaired F-actin dynamics result in mislocalization of the mitochondrial fission protein Drp1from mitochondria, leading to abnormal mitochondrial morphology and function in our α-synuclein transgenic fly model [39]. Since normal mitophagosome formation requires preserved mitochondrial dynamics [22,84], it is possible that excess F-actin stabilization impairs mitophagy through multiple mechanisms in α-synucleinopathy.

These findings are particularly interesting because gene products of two intensively studied autosomal recessive Parkinson's disease loci, parkin and PINK1, play key roles in mitophagy [22]. Thus, our findings provide a mechanistic link between common forms of Parkinson's disease related to α-synuclein aggregation and deposition and rarer forms of the disorder linked specifically to mitophagy defects. The more widespread pathology in the broader range of α-synucleinopathies, including dementia with Lewy bodies and multiple system atrophy, compared with parkinsonism caused by mutations in the genes encoding parkin and PINK1, might reflect additional cellular targets of excess actin stabilization. Our current work suggests that autophagosomes may represent one such target. Therapeutic strategies that reduce the deleterious F-actin stabilization caused by α-synuclein may have beneficial effects on multiple downstream targets in sporadic Parkinson's disease, dementia with Lewy bodies, and multiple system atrophy.

Certainly, autophagy itself is an attractive therapeutic target in α-synucleinopathies and other neurodegenerative diseases characterized by abnormal protein aggregation [85,86]. In rodent α-synucleinopathy models, promoting autophagy by expressing BECN1 [87,88], ATG7 [70], LAMP2A [89,90] or treatment with rapamycin [70] reduces α-synuclein levels and ameliorates pathological disease markers. Since α-synuclein itself is a substrate of autophagy [24,25], promoting autophagy should reduce α-synuclein levels, including toxic phosphorylated [44,91,92], oligomeric [45], and larger α-synuclein inclusions. Activating autophagy would interrupt a deleterious feedforward loop and promote autophagosome clearance [28] and transport [93]. Collectively, our findings provide mechanistic insight into the basis of autophagic and mitophagic dysfunction in α-synucleinopathies, including Parkinson's disease, and suggest new potential therapeutic strategies in these common and currently untreatable disorders.

## Materials and methods

### *Drosophila* genetics

*Drosophila* crosses were performed at 25˚C. All flies were aged at 25˚C for 10 days unless otherwise noted in the figure legend. Equal numbers of male and female flies were used in each

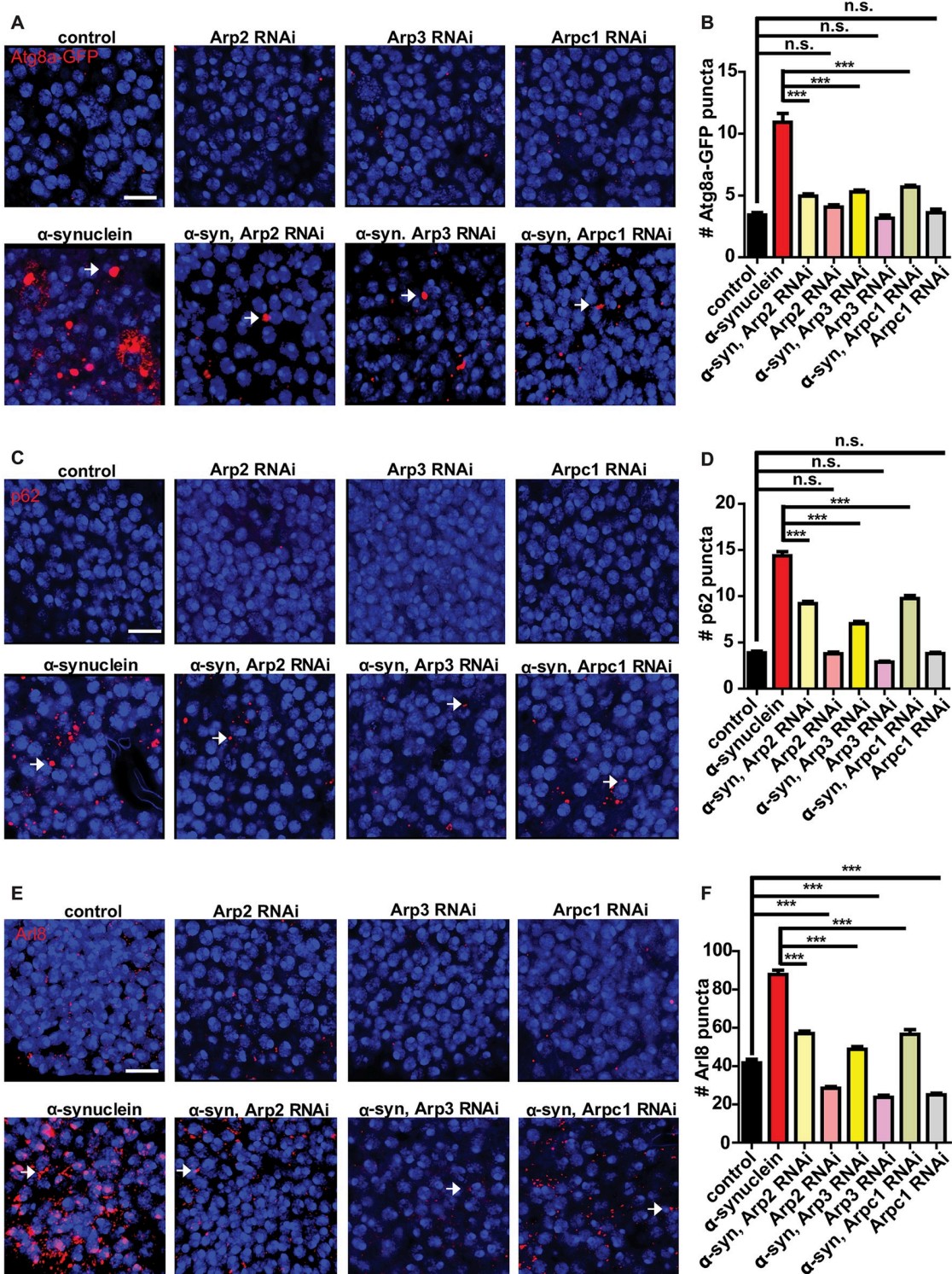

**Fig 8. Inhibition of the Arp2/3 actin nucleation pathway rescues the effect of α-synuclein on autophagosomes and lysosomes.**
(A) Representative images of Atg8a-GFP (arrows) following knockdown of Arp2, Arp3 or Arpc1in the presence and absence of α-synuclein. (B) Quantification of Atg8a-GFP-positive puncta shows a reduced number of Atg8a puncta in α-synuclein transgenic flies. (C) Representative images of p62 puncta (arrows) in flies with reduced Arp2, Arp3, or Arpc1 in presence and absence of α-synuclein. (D) Quantification of p62-positive puncta shows Arp2, Arp3, or Arpc1 knockdown reduced number of p62 puncta in α-synuclein transgenic flies. (E) Representative images of Arl8 (arrows) in flies with knockdown of Arp2, Arp3, or Arpc1 in presence

and absence of α-synuclein. (F) Quantification of Arl8-positive puncta shows Arp2, Arp3, or Arpc1 knockdown reduced number of Arl8 positive structures in α-synuclein transgenic flies. Control genotype in (A,B): *nSyb-QF2, nSyb-GAL4, UAS-Atg8a-GFP / +*. Control genotype in (C-F): *nSyb-QF2, nSyb-GAL4/+*. Full genotypes are provided for all animals in S1 Text. *** p<0.001, ANOVA followed by Tukey's multiple comparisons test. Data are represented as mean ± SEM. n = 6 per genotype. The number of puncta per 500 μm$^2$ is presented. Scale bars are 5 μm. Flies are 10 days old.

experiment. The pan-neuronal drivers *nSyb-GAL4* and *nSyb-QF2* were used for all experiments. Our laboratory has previously described *QUAS-wild type α-synuclein* [39] and *UAS-Gelsolin* [40] transgenic flies. The following stocks were obtained from the Bloomington *Drosophila* Stock Center: *Rab2$^{EYFP}$, UAS-cofilin/tsr, UAS-cofilin$^{S3A}$, UAS-cofilin$^{S3E}$, nSyb-GAL4, UAS-GFP-mCherry-Atg8a, Arp2$^{f04069}$, UAS-Arp2$^{JF02785}$, UAS-Arp3$^{HMS0071}$* (Arp3 RNAi #1), *UAS-Arp3$^{HMJ21357}$* (Arp3 RNAi #2), *UAS-Arpc1$^{JF01763}$, nSyb-GAL4, elav-GAL4*. The following stocks were kind gifts of the indicated investigators: *UAS-Atg8a-GFP*, Thomas Neufeld; *nSyb-QF2*, Christopher Potter.

## Histology, immunohistochemistry and imaging

*Drosophila* heads were fixed in formalin and embedded in paraffin before sectioning. Serial frontal sections (2 or 4 μm) of the entire brain were prepared and mounted on glass slides. To assess neuronal density, sections were stained with hematoxylin and imaged using the imaging software SPOT. The number of anterior medulla neurons in an approximately 1000 μm$^2$ area were counted using the cell counter plugin in ImageJ.

For immunostaining on paraffin sections, antigen retrieval was performed by microwaving slides in sodium citrate buffer for 15 minutes. Slides were then blocked in 2% milk in PBS with 0.3% Triton X-100, followed by overnight incubation with primary antibodies at room temperature. Primary antibodies to the following proteins were used at the indicated concentrations: α-synuclein (1:10,000, CA00944, kind gift of BioLegend), Arl8 (1:1,000, Developmental Studies Hybridoma Bank), GABARAP (1:1,000, endogenous Atg8a, E1J4E, Cell Signaling), GFP (1:1,000, N86/8, Neuromab; 1:2,000), p62 (ref(2)P) (1:1,000, this study), tyrosine hydroxylase (1:500, Immunostar), ATP5A (1:1,000, 15H4C4, Invitrogen). After three washes with PBS-Triton, slides were incubated with appropriate secondary antibodies coupled to Alexa Fluor 488, Alexa Fluor 555, or Alexa Fluor 647. Rabbit polyclonal antibodies directed against *Drosophila* p62 (ref(2)P) were created by Covance using the peptide sequence PRTEDPVTTPRSTQ. Quantification of dopaminergic neurons on paraffin sections was performed by immunostaining with biotinylated secondary antibodies, followed by avidin–biotin–peroxidase complex (Vectastain Elite; Vector Laboratories), as previously described [43].

Imaging of fluorescent markers was performed with Zeiss laser-scanning confocal microscopy. To visualize morphology of lysosomal compartments, fresh whole mount brains from 10-day-old flies were incubated with 0.05 μM LysoTracker for 5 minutes at room temperature, mounted in PBS, and imaged immediately. The number of puncta per 500 μm$^2$ was counted. To assess autophagic flux, freshly dissected brains from 10-day-old animals were imaged using Airyscan. All images were taken using 63X-oil lens with 1.3X zoom and processed in Zen Blue 2.6. GFP-mCherry-Atg8a structures were analyzed as previously described [58]. Briefly, images were opened in ImageJ and the intensity of GFP and mCherry was measured in 50 regions of interest (ROI) from each animal. For surface intensity plot, a ROI was drawn, and then the intensity of GFP and mCherry was plotted along the selected ROI. Each image shown is a maximal projection of a Z-stack consisting of 8 stacks of 0.2 μm thickness. During acquisition, the laser intensity was the same across all genotypes, and each channel was tested for saturation across all stacks. A sequential scan was performed to avoid any crosstalk between fluorophores.

Post-acquisition, the intensity of all the images were modified in an identical manner across genotypes to identify low intensity positive signals in the control groups.

Quantification of mitophagy using the mt-Keima reporter was performed as described [65]. Briefly, an area of interest was defined, and the emission at 543 nm and 458 nm was recorded. The 543 nm/458 nm ratio was calculated and expressed per unit area. For colocalization studies, the Pearson coefficient was calculated from 3–4 images per animal, each image containing 4–10 cells. For each animal, the average Pearson coefficient from the 3–4 images were plotted.

## Electron microscopy

For mitochondrial morphology, brains from 20-day-old α-synuclein transgenic and control flies were dissected out of the cuticle and fixed in 2% paraformaldehyde and 2.5% glutaraldehyde (Polysciences). Following incubation in 1% osmium tetroxide (Electron Microscopy Sciences) in 1.5% potassium ferrocyanide (MP Biomedicals) for 1 hour, fly brains were incubated in 1% uranyl acetate for 30 minutes, and processed through 70, 90, and 100% ethanol solutions. Brains were then incubated in propylene oxide for 1 hour, embedded in Epon/Araldite mixture, and allowed to polymerize for 2 days at 65˚C. Thin sections were cut and examined with a conventional Tecnai G2 Spirit BioTWIN transmission electron microscope.

## Immunoblotting

For western blot analysis, *Drosophila* heads were homogenized in 15 μl of 2x Laemmli buffer (Sigma-Aldrich). Samples were boiled for ten minutes and subjected to SDS-PAGE. Protein were transferred onto nitrocellulose membranes (Bio-Rad), blocked in 2% milk in PBS with 0.05% Tween 20 (PBSTw), and incubated overnight at 4˚C with primary antibodies. Membranes were washed three times with PBSTw and incubated for three hours with appropriate HRP-conjugated secondary antibody (SouthernBiotech) at room temperature. Following multiple washes with PBSTw, proteins were visualized by enhanced chemiluminescence (Alpha Innotech). Protein transfer and loading were monitored by Ponceau S staining and by reprobing blots with an antibody directed to GAPDH. All immunoblots were repeated at least three times. Primary antibodies to the following proteins were used at the indicated concentrations: α-synuclein (1:6,000,000, H3C, Developmental Studies Hybridoma Bank), GAPDH (1:40,000, Invitrogen), cofilin (1:1,000,000, Anna Marie Sokac).

## Measurement of oxygen consumption and extracellular acidification rates

The oxygen consumption rate (OCR) and extracellular acidification rate (ECAR) were measured using a Seahorse XF96 metabolic analyzer following the procedure recommended by the manufacturer. Briefly, for all the experiments, brains from 10-day-old flies of all genotypes were dissected and plated at one brain per well on XF96 plates (Seahorse Bioscience), and metabolic parameters assayed as described [66]. The OCR values were normalized to DNA content using a CyQUANT assay (ThermoFisher) following the manufacturer's protocol.

## Statistical analyses

The sample size (n), mean and SEM are indicated in the figure legends. All statistical analyses were performed as described in the figure legends using GraphPad Prism 5.0. For comparisons across more than 2 groups, one-way ANOVA with Tukey post-hoc analysis was used. For comparison of 2 groups Student's t-tests were performed. P values are indicated in the figure legends.

## Supporting information

**S1 Fig. α-synuclein transgenic fly brains show progressive age-dependent alterations of the autophagosomal system.** (A) Representative GFP-immunofluorescence images of the anterior medulla and (B) quantification of Atg8a-GFP puncta. Control genotype in (A,B): *UAS-Atg8a-GFP/ nSyb-QF2, nSyb-GAL4.* (C) Immunofluorescence staining of endogenous Atg8a and (D) quantification of Atg8a-positive puncta. (E, F) Age-dependent accumulation of p62-immuno-reactive aggregates as demonstrated by immunofluorescence (E) and (F) quantification of p62-postive puncta. Control genotype in (C-F): *nSyb-QF2, nSyb-GAL4/+.* Full genotypes are provided for all animals in S1 Text. *** p<0.0001, ANOVA with Tukey's multiple comparisons test. n.s. not significant. Data are represented as mean ± SEM. n = 6 per genotype. Scale bars are 5 μm. The number of puncta per 500 μm² is presented. The ages of the flies are indicated in the figure labels.
(TIF)

**S2 Fig. Expression of EGFP or ß-galactosidase does not alter autophagosome or lysosome markers.** (A) Schematic diagram illustrating the analyzed *Drosophila* brain region of the anterior medulla with inset showing a hematoxylin-stained histological section from the same area. (B,C) No change in the numbers of Atg8a-positive puncta in brains of flies expressing EGFP or ß-galactosidase. (D,E) No change in the numbers of p62-immunoreactive aggregates in brains of flies expressing EGFP or ß-galactosidase. (F,G) No change in the numbers of Arl8-immunoreactive puncta in brains of flies expressing EGFP or ß-galactosidase. n = 6 per genotype. (H) Total number of GFP-mCherry-Atg8a-positive puncta in flies with and without expression of ß-galactosidase. (I) Ratio of GFP to mCherry fluorescence in brains of flies with and without expression of ß-galactosidase. Control genotype in (B,C): *UAS-Atg8a-GFP/ nSyb-QF2, nSyb-GAL4.* Control genotype in (D-G): *nSyb-QF2, nSyb-GAL4/+.* Full genotypes are provided for all animals in S1 Text. No significant differences were detected (p>0.05, ANOVA with Tukey's multiple comparisons test in B-G, t-test in I). Data are represented as mean ± SEM. n = 6 per genotype in (B-G). n = 4 per genotype in (H,I). Scale bars in (B,D,F) are 2 μm and 5 μm in (H). The number of puncta per 500 μm² is presented in (C,E,G). Flies are 10 days old.
(EPS)

**S3 Fig. Inactive cofilin does not rescue α-synuclein-induced increase in Atg8a-positive autophagosomes.** (A) Expression of wild type (WT) and constitutively active cofilin[S3A], but not inactive cofilin[S3E], rescues α-synuclein-mediated increases in Atg8a-GFP-positive puncta in α-synuclein transgenic flies. (B) Quantification of puncta from (A). (C) Immunoblot of *Drosophila* head homogenates showing no change in α-synuclein levels among α-synuclein transgenic flies with and without cofilin transgene expression. (D) Immunoblot of *Drosophila* head homogenates showing comparable cofilin expression levels among all three forms of cofilin. (E) Immunoblot of *Drosophila* head homogenates showing no change in α-synuclein levels between α-synuclein transgenic flies with and without gelsolin overexpression. All blots were reprobed for GAPDH to illustrate equivalent protein loading. Control genotype in (A,B): *nSyb-QF2, nSyb-GAL4, UAS-Atg8a-GFP / +.* Control genotype in (C-E): *nSyb-QF2, nSyb-GAL4 / +.* Full genotypes are provided for all animals in S1 Text. *** p<0.0001, ANOVA with Tukey's multiple comparisons test. n = 6 per genotype in (A,B). n = 3 per genotype in (C-E). Scale bar is 5 μm (A). The number of puncta per 500 μm² is presented (A). Flies are 10 days old in (A,B). Flies are 1–3 days old in (C-E).
(TIF)

**S4 Fig. Gelsolin or cofilin overexpression does not alter autophagosomes or lysosomes in the absence of transgenic α-synuclein.** (A-H) No change in the number of Atg8a-GFP (A

arrows, B), p62 (C arrows, D), Arl8 (E arrows, F), or LysoTracker (G arrows, H) -positive puncta in flies expressing gelsolin or cofilin. (I) Representative images of brains from flies expressing GFP-mCherry-Atg8a with and without gelsolin or cofilin show no change in the number of mCherry (arrows) or GFP-positive puncta, as quantified by the number of GFP-mCherry-Atg8a dual-positive puncta in (J). Control genotype in (A,B): *UAS-Atg8a-GFP/ nSyb-QF2, nSyb-GAL4*. Control genotype in (C-H): *nSyb-QF2, nSyb-GAL4/+*. Control genotype in (I,J): *UAS-GFP-mCherry-Atg8a/+; nSyb-QF2, nSyb-GAL4/ +*. Full genotypes are provided for all animals in S1 Text. Data are represented as mean ± SEM. n = 6 per genotype in (A-H). n = 4 per genotype in (I,J). Scale bars are 5 μm. The number of puncta per 500 μm² are presented in (B,D,F,H) and per 1000 μm² in (J). Flies are 10 days old.
(TIF)

**S5 Fig. Gelsolin or cofilin overexpression does not alter mitophagy in the absence of transgenic α-synuclein.** (A) Overlay of live mt-Keima emission at 543 (red) and 458 (green) showing no change in high 543/458 puncta (arrows) in brains of flies overexpressing gelsolin or cofilin in the absence of human α-synuclein. (B) 543/458 ratio area/total mitochondrial area was quantified as an index of mitophagy in control and gelsolin or cofilin transgenic flies and does not show a statistically different alteration (p = 0.54 for gelsolin, p = 0.77 for cofilin, ANOVA with Tukey's multiple comparisons test). Control genotype: *UAS-mt-Keima / +; nSyb-QF2, nSyb-GAL4/ +*. Full genotypes are provided for all animals in S1 Text. Data are represented as mean ± SEM. n = 4 per genotype. Scale bar is 5 μm (A). Flies are 10 days old.
(EPS)

**S6 Fig. Effects of gelsolin or cofilin overexpression on bioenergetics in the absence of transgenic α-synuclein.** (A-G) Metabolic profiling of whole *Drosophila* brains in Seahorse XFe96-well culture microplates (Agilent) assessing oxygen consumption rate (OCR, A), extracellular acidification rate (ECAR, B), proton efflux rate (PER, C), non-mitochondrial respiration (D), basal respiration (E), maximal respiration (F), and proton leak (G) in control (not expressing human α-synuclein) flies with overexpression of gelsolin or cofilin. Control genotype: *nSyb-QF2, nSyb-GAL4/+*. Full genotypes are provided for all animals in S1 Text. ** p<0.01, *** p<0.001, ANOVA with Tukey's multiple comparisons test. Data are represented as mean ± SEM. n = 6 per genotype. Flies are 10 days old.
(EPS)

**S7 Fig. Inhibition of Arp2/3 actin nucleation pathway reduces α-synuclein-induced increase in autophagic defects.** (A) Representative images of Atg8a-GFP (arrows) following knockdown of Arp2 or Arp3 using a heterozygous loss of function Arp2 allele (*Arp2^f04069^ / +)* or a confirmatory second Arp3 RNAi line. (B) Quantification of Atg8a-GFP-positive puncta shows a reduced number of Atg8a puncta in α-synuclein transgenic flies. n = 6 per genotype. The solid line in (B) indicates the control value (genotype: *nSyb-QF2, nSyb-GAL4, UAS-Atg8a-GFP / +*). (C) Immunoblot of *Drosophila* head homogenates showing no change in α-synuclein levels among α-synuclein transgenic flies with and without Arp2/3 complex member knockdown. The blot is reprobed for GAPDH to illustrate equivalent protein loading. Control genotype in (C): *nSyb-QF2, nSyb-GAL4 / +*. Full genotypes are provided for all animals in S1 Text. *** p<0.001, ANOVA followed by Tukey's multiple comparisons test. Data are represented as mean ± SEM. n = 6 per genotype in (A,B) and 3 per genotype in (C). Scale bar is 5 μm (A). The number of puncta per 500 μm² is presented (B). Flies are 10 days old in (A,B) and 1–3 days old in (C).
(TIF)

**S1 Video. Colocalization of mitochondria with enlarged autophagosomes in α-synuclein transgenic fly neurons.** Representative super-resolution image of neurons from an

α-synuclein transgenic fly brain showing colocalization of mitochondria (ATP5A, purple) with enlarged autophagosomes marked by transgenic Atg8a-GFP (yellow) with 360-degree rotation along the X-axis of a Z-stack.
(MP4)

**S1 Text. Genotypes for all experiments.**
(DOC)

## Acknowledgments

Fly stocks were obtained from the Bloomington *Drosophila* Stock Center (NIH P40-OD018537) and Drs. T. Neufeld and C. Potter. The anti-cofilin antibody was the kind gift of Dr. A.M. Sokac. Kit Tuen provided excellent technical assistance. Monoclonal antibodies were obtained from the Developmental Studies Hybridoma Bank developed under the auspices of the NICHD and maintained by the University of Iowa, Department of Biology, Iowa City, IA 52242, and the UC Davis/NIH NeuroMab Facility.

## Author Contributions

**Conceptualization:** Souvarish Sarkar, Mel B. Feany.

**Formal analysis:** Souvarish Sarkar.

**Funding acquisition:** Mel B. Feany.

**Investigation:** Souvarish Sarkar, Katja Sygnecka, Mel B. Feany.

**Supervision:** Mel B. Feany.

**Writing – original draft:** Katja Sygnecka, Mel B. Feany.

**Writing – review & editing:** Souvarish Sarkar, Abby L. Olsen, Kelly M. Lohr, Mel B. Feany.

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
