## [Decision Letter · Decision Letter 0]

4 Jan 2021

Dear Dr Feany,

Thank you very much for submitting your Research Article entitled 'α-synuclein impairs autophagosome maturation through abnormal actin stabilization' to PLOS Genetics.

The manuscript was fully evaluated at the editorial level and by two independent peer reviewers. The reviewers appreciated the attention to an important topic but identified some concerns that we ask you address in a revised manuscript.  You should be able to address their concerns without further experimentation (unless the GFP saturation issue raised by Reviewer #1 require re-imaging) and we should be able to make an editorial decision on the revised manuscript without additional external review.

We therefore ask you to modify the manuscript according to the review recommendations. Your revisions should address the specific points made by each reviewer.

[LINK]

Yours sincerely,

Gregory P. Copenhaver

Editor-in-Chief

PLOS Genetics

Reviewer's Responses to Questions

**Comments to the Authors:**

Reviewer #1: The manuscript explores the possibility that the expression of human alpha-synuclein stabilizes F-actin and a consequence impairs autophagosome maturation which in turn promotes dopaminergic cell loss.

To test this hypothesis they co-express with alpha-synuclein severing proteins gelsolin and cofilin in order to reduce F-actin levels in the cell. the expression of these two proteins revert several markers associated with the reduction of autophagosome activity.

The manuscript is clearly written and should be published however there are two minor concerns that should be addressed before publication.

1) Labels in immunofluorescent panels figure 1 are hard to read. Figure 1 images are of very low quality.

2) Green signal in figure 2 is overly saturated, why? What were the acquisition conditions? It appears that there is practically no dynamic range in the green channel.

Reviewer #2: Summary: The manuscript by Sarkar et al., is a follow up study based on the development of a well-validated Drosophila model (Ordonez et al., Neuron, 2018) for �-synuclein neuropathies including the formation of so-called Lewy bodies. In that previous study, the Feany lab reported that reduced mitochondrial function was linked to abnormal actin-mediated processes but the mechanisms underlying these defects remained to be established. Cellular defects characteristic of the human neuropathies were reproduced in Drosophila neurons expressing a disease-inducing version of the human �-synuclein gene. In the current manuscript, the authors take a significant step forward in delineating the mechanism underlying this actin-based pathology by identifying a stabilizing role of the actin cytoskeleton in blocking a key step in autophagy. They use a range of well crafted tools including antibody staining, expression of compartment-specific markers (some of which change output in response to differing cellular compartments), and over-expression of actin severing proteins (gelsolin or cofilin) to analyze in detail the process of autophagy in neurons expressing the disease associated form of �-synuclein. The authors find that the autophagy, in particular the fusion of autophagosomes to the lysosomal compartment, is greatly reduced in neurons of �-synuclein flies and show further that reduction of key elements of the actin remodeling system (Arp2,3) can greatly ameliorate this condition at the cellular level. These results point the way to potential future therapies (e.g., with rapamycin) for treating �-synucleinopathies by targeted clearing of actin-tethered autophagosomes, potentially by promoting autophagic clearance of �-synuclein itself.

Critique: Experiments presented in this study are of high quality and support the authors’ conclusions and interpretations of the data. These findings of are of broad interest and represent a substantial advance in the field by delineating an actin-dependent block in autophagy as the key cellular defect in response to expression of a disease associated form of �-synuclein. Based on the quality of the work and its impact, I believe this manuscript merits publication in PLOS Genetics and recommend publishing this paper without any additional required experiments. A few suggestions that the authors may wish to consider in formulating the final version of the manuscript are included below:

1) An interesting observation that the authors make in the context of examining the rescuing effects of over-expressing actin severing proteins is that the size of Atg8a-GFP expressing vesicles is decreased with only minor if any effects on the number of these structures (Fig. 3A). The effect is also seen to some degree with the p62 and Arl8 markers (Fig. 3C,E). The authors note this effect in the text and comment reasonably that experimental analysis of the basis for this primary effect on autophagosome size rather than number is beyond the scope of the current study. I agree that no new experiments need to be performed, however, it might be helpful for the reader to understand what the possible basis for this specific effect could be. For example, might this finding suggest that rather than nucleating the initial formation of excess autophagosomes the actin-dependent effect of �-synuclein is primarily involved in a feedback mechanism that keeps adding membrane to nascent autophagosomes (perhaps a non-actin-dependent mechanism triggers their initial formation?). Is such a two-step self-amplifying model tenable and if so how might the action of the disease form of �-synuclein be involved? Also, could there be a connection between the formation of the cellular inclusions (e.g., shown in Fig. 1I) and such a positive feedback cycle? Could accretion of such a mass of complex membrane impede fusion with the lysosomal compartment?

2) The authors show convincing evidence that RNAi of components of the Arp2,3 complex greatly reduce the cellular effects of �-synuclein. Does this cellular rescue also translate to rescue in longevity studies (or other organism level phenotypes)? If not, perhaps the authors could state that and offer a potential explanation?

3) Also regarding the suppression of the �-synuclein phenotype by reduction in Arp2,3 complex members, does heterozygosity for mutant loss-of-function alleles of these same genes (Arp2, Arp3 Arpc1) also result in suppression of cellular �-synuclein phenotypes? Along the same lines, is there any evidence from GWAS studies in humans that heterozygosity for Arp2,3 components reduce the probability of developing �-synucleinopathies or that amplification of any these loci (e.g., by copy number variation or local duplications) score as a risk factor?

4) A small stylistic point. In my view, it is best not to begin sentences with a preposition (e.g., To determine how….). I know that this style has now become accepted practice, but I personally find it to be a repetitious form that can readily be avoided. For example, the following more active rephrasing would be preferable in my humble opinion:

Current sentence: To detect autophagosomes, we crossed flies containing the transgenic reporter construct UAS-Atg8a-GFP to α-synuclein transgenic flies and visualized tagged Atg8a by immunofluorescence in brain sections.

Alternative phrasing: We detected autophagosomes by crossing flies containing the transgenic reporter construct UAS-Atg8a-GFP to α-synuclein transgenic flies and visualized tagged Atg8a by immunofluorescence in brain sections.

**Have all data underlying the figures and results presented in the manuscript been provided?**

Reviewer #1: Yes

Reviewer #2: Yes

PLOS authors have the option to publish the peer review history of their article (what does this mean?). If published, this will include your full peer review and any attached files.

Reviewer #1: No

Reviewer #2: No

---

## [Editor Report · Decision Letter 1]

14 Jan 2021

Dear Dr Feany,

We are pleased to inform you that your manuscript entitled "α-synuclein impairs autophagosome maturation through abnormal actin stabilization" has been editorially accepted for publication in PLOS Genetics. Congratulations!

Yours sincerely,

Gregory P. Copenhaver

Editor-in-Chief

PLOS Genetics

Comments from the reviewers (if applicable):

**Data Deposition**

http://datadryad.org/submit?journalID=pgenetics&manu=PGENETICS-D-20-01755R1

**Press Queries**

---

## [Editor Report · Acceptance letter]

2 Feb 2021

PGENETICS-D-20-01755R1 

α-synuclein impairs autophagosome maturation through abnormal actin stabilization 

Dear Dr Feany, 

We are pleased to inform you that your manuscript entitled "α-synuclein impairs autophagosome maturation through abnormal actin stabilization" has been formally accepted for publication in PLOS Genetics! Your manuscript is now with our production department and you will be notified of the publication date in due course.

With kind regards,

Alice Ellingham

PLOS Genetics

On behalf of:
